# Effectiveness of Cover Crops to Reduce Loss of Soil Organic Matter in a Rainfed Vineyard

**Manuel López-Vicente [1,\*], Elena Calvo-Seas [2], Sara Álvarez [3] and Artemi Cerdà [4]**

[1] Team Soil, Water and Land Use, Wageningen Environmental Research, Droevendaalsesteeg 3, 6708RC Wageningen, The Netherlands

[2] Geographer, Independent Scholar, Casa Calvo, 22144 Bierge (Huesca), Spain; eleseas@gmail.com

[3] Unit of Woody and Horticultural Crops, Technological Agriculture Institute of Castilla y León, ITACyL, 47071 Valladolid, Spain; alvmarsa@itacyl.es

[4] Soil Erosion and Degradation Research Group, Department of Geography, Valencia University, Blasco Ibàñez 28, 46010 Valencia, Spain; artemio.cerda@uv.es

\* Correspondence: mlopezvicente@gmail.com

**Abstract:** Cover crops (CCs) minimize the loss of soil in permanent cropping systems where the soil is usually bare due to intense tillage or overuse of herbicides. The topsoil, the richer layer in soil organic carbon and organic matter (OM), is affected by water erosion. Nature-based solutions appear as a suitable option for sustainable farming. In this study, the effectiveness of two years of CC management to reduce the OM loss is evaluated in a rainfed vineyard in a rolling landscape (Huesca, NE Spain). Two sediment traps collected runoff over 15 months. Topsoil OM contents (1.64% and 1.60%) and sediment/soil OM enrichment ratio (2.61 and 3.07) were similar. However, the average annual rate of OM loss was 3.6 times higher in the plot with lower vegetation cover than in the plot with CCs (1.29 vs. 0.35 $kg_{OM}$ $ha^{-1}$ $yr^{-1}$). The concentration of $OM_{Sed}$ showed a negative relationship with the net soil loss; and significant differences appeared between the $OM_{Sed}$ in the months with low and moderate-to-high ground cover. CCs are an excellent nature-based solution to control the unsustainable soil and OM losses measured in vineyards, which will contribute to achieve the Sustainable Development Goals of the United Nations.

**Keywords:** cover crop; vineyard; soil erosion; soil organic matter; enrichment ratio; sediment trap; Mediterranean climate

## 1. Introduction

In 2015, all United Nations Member States adopted the resolution entitled "Transforming Our World: The 2030 Agenda for Sustainable Development: [1]. This Agenda encompasses the 17 Sustainable Development Goals (SDGs) and 169 targets with global ambitious scale. Land degradation neutrality (LDN) is the concept behind the target 15.3, which is part of the SDG 15 called "Life on Land." The concept of LDN aims to respond to "the need for urgent action to reserve land degradation and achieve a land-degradation neutral world" [2]. In croplands, sustainable solutions need to embed short-term management of the soil–water–plant system in long-term landscape planning [3]. Achieving the target of land degradation neutrality would decrease the environmental footprint of agriculture, while supporting food security and sustaining human wellbeing [4]. The soil system acts as a key component of the Earth system by regulating the hydrological, erosional, and biogeochemical cycles [5]. Therefore, the soil is where we first need to act to control the non-sustainable management and create enabling conditions to transition toward sustainable land management [6]. This is especially true for agricultural soils, where soil loss rates are commonly unsustainable and above those considered as tolerable [7–9].

Soil erosion, by water and wind, is among the most serious threats to preserve soil and water resources at any spatial and temporal scale [10,11]. The expansion of agriculture and livestock farming has resulted in a significant increment of soil erosion rates that explain the development of erosional landscapes and sedimentary structures (recent alluvial plains, alluvial fans, deltas, and flat valleys infilled of sediment) in many countries, such as in Spain [12]. Within the European agricultural areas (mean estimated rate of soil loss of 4.21 Mg ha$^{-1}$ yr$^{-1}$), soils of the permanent crops (e.g., vineyards, olive groves, fruit tree orchards) have much higher soil losses (9.47 Mg ha$^{-1}$ yr$^{-1}$) than arable land (2.67 Mg ha$^{-1}$ yr$^{-1}$) and pastures (2.02 Mg ha$^{-1}$ yr$^{-1}$) [13]. This fact is explained because most of the vineyards and olive trees are located in hilly Mediterranean areas with high rainfall erosivity and due to the unsustainable tillage (bare soil) and exploitation (new large fields) management practices. It is well accepted by the scientific community that the traditional rainfed and tilled agriculture land in the Mediterranean contributes to high erosion rates. This was assessed by means of direct measurements (plots and watersheds) and rainfall simulator experiments along the last three decades. Gómez et al. [14] measured in Córdoba (Andalucía, South Iberian Peninsula) 4 Mg ha$^{-1}$ yr$^{-1}$ of soil loss on plots of 6 × 12 m and a runoff coefficient—the ratio between the amount of water that leaves (via runoff) from a hydrological unit (measured in mm or L m$^{-2}$) and the amount of water that enters (via rainfall) into the same unit (in mm or L·m$^{-2}$)—of 7.4% under tillage. The use of herbicides increased the soil losses to 8.5 Mg ha$^{-1}$ yr$^{-1}$ and 21.5%, respectively. Other vineyards also showed high erosion rates; Novara et al. [15] found 9.5 Mg ha$^{-1}$ yr$^{-1}$ under tillage management in Sicily. This is confirmed with the data generated with rainfall simulators by Rodrigo-Comino et al. [16] that show non-sustainable soil losses in the vineyards in eastern Spain. Similar soil erosion rates were found in other Mediterranean crops such as almonds [17], apricots [18], and flood irrigated persimmon orchards [9].

Among the distinct measures to minimize soil erosion in permanent crops, the use of ground cover plants (e.g., spontaneous vegetation, cover crops, pruning remains, straw mulch) has been proved as an effective and environmentally friendly alternative [19]. Homogeneous and mixed temporary cover crops (CCs) favor surface water infiltration [20,21] and reduce runoff generation compared with fields managed under conventional tillage (CT)—bare soil in the inter-row areas due to ploughing and/or use of herbicides [22]. The use of weeds as cover crops was found to be a good policy in citrus orchards due to the fact that they reduce the soil losses from 3.8 to 0.7 Mg ha$^{-1}$ h$^{-1}$ [23]. Cover crops in vineyards also contribute to reduce soil losses; Ruiz-Colmenero et al. [24] found that tillage contribute with 5.88 Mg ha$^{-1}$ yr$^{-1}$ and a cover of *Brachypodium* sp. reached 0.78 and *Secale* 1.27 Mg ha$^{-1}$ yr$^{-1}$.

Next to reducing soil erosion, which prevents also the export of carbon from the fields, certain soil management strategies can increase the amount of carbon in the soil and sequester carbon for improved soil health and climate mitigation. For example, Novara et al. [8] found increments of soil organic carbon (SOC) up to 6% and 9% in Italian vineyards with CC in flat and sloping areas, respectively, and compared with vineyards under CT. In central Spain, under semi-arid conditions and only three years after ground cover establishment, García-Díaz et al. [25] found that soil organic carbon stocks increased up to 1.62 and 3.18 Mg ha$^{-1}$ in the vineyards with seeded CC and spontaneous vegetation, respectively, compared to the CT vineyard. In a recent meta-analysis in Mediterranean permanent crops, Vicente-Vicente et al. [26] observed that soil carbon sequestration potential in olive farming was much higher than in vineyards, mainly due to the relative large space of the field that is the beneficiary of the good management practices, such as ground cover plants or organic amendments. In a review study in commercial vineyards in southern Spain, Guzmán et al. [27] identified a significant improvement in the SOC content and soil aggregate stability in the vineyards with ground covers in comparison with the bare soil vineyards. However, these authors observed a large variability among vineyards that precludes the identification of other impacts and differences among the different kind of temporary plants (plant species of the cover crops and spontaneous vegetation) followed by the winegrowers. Processes of carbon transfer and sequestration in Grenache vineyards, which is a large and economically important variety in Mediterranean vineyards [28], have been rarely studied. Capó-Bauçà et al. [29] investigated the long-term effects of natural green cover, established over seven

years, on the soil physicochemical characteristics in an experimental Grenache variety vineyard located in Mallorca (Spain). These authors found that natural weed cover resulted in an increment of 1% of the organic carbon content, larger dry aggregate sizes, a reduction of 0.36 g·cm$^{-3}$ of bulk density, and higher functional microbial community diversity. After 21 years, Novara et al. [30] measured an increase of 61 Mg $CO_2$ ha$^{-1}$ of soil C pools due to the organic farming applied to an orange plantation in the orchards of Valencia region in Eastern Spain. This was due to the chipped pruned branches, the weed cover and the use of 5000 Kg ha$^{-1}$ of manure annually.

Although data is already available on soil organic matter recovery with alternative management to tillage and herbicides, there is a need to diversify the research on different types of crops, management strategies, climatic regions, and physiographic conditions. This will enable farmers to obtain healthy soil health and contribute to achieving the SDGs. The topography and roughness of the land determine surface runoff and consequent sediment delivery. However, this factor is not studied at plot scale and in detail. The effectiveness of cover crops to reduce soil organic matter loss in vineyards under different topographical conditions is a relevant issue as vineyards are mostly located in rugged terrain and steep slopes where soils have poor fertility and are highly erodible. The review carried out by Rodrigo-Comino [31] demonstrates that soil erosion is high in vineyards around the world, where the organic-rich topsoil is especially vulnerable to be lost from the fields. This organic-rich material is translocated from the fields where it is eroded to depositional sites. This process can be explained by the sediment/soil organic matter (OM) enrichment ratios (ER$_{OM}$) [32]. In semi-arid croplands of Spain, Boix-Fayos et al. [33] estimated an average organic carbon ER at sub-catchment scale (8–125 ha) of 0.59 ± 0.43. Although the study of carbon and organic matter ER is not rare in olive groves [34] and other croplands [35], the assessment of ER$_{OM}$ in vineyards has received little attention (e.g., Ruiz-Colmenero et al. [36]; ER$_{OC}$ ranging from 1.4 to 2).

In this study, we hypothesized that the OM loss is influenced by the local changes of the soil surface characteristics—regarding surface coverage, slope gradient, and total drainage area—within the same vineyard managed with a seeded cover crop. Most previous studies have dealt with this issue at plot or field scale. The novel aspect of this study is the assessment on how the internal changes within the same field influence the magnitude of the benefits of CC on soil and carbon loss reduction. To achieve this goal, a rainfed vineyard was selected in a rolling landscape under Mediterranean conditions and managed with a mixed cover crop of common sainfoin (forage legume) and spontaneous vegetation. Topographic and ground characteristics were assessed at high spatial resolution; using drone-derived imagery, two sediment traps were installed, and soil and sediment organic matter contents were measured during a 15-month test period. This study aims to better understand the actual effectiveness of cover crops to reduce organic matter loss in vineyards under natural (non-controlled) conditions where ground cover plant development presents high spatial heterogeneity.

## 2. Materials and Methods

### 2.1. Study Area and Sediment Traps

The study area is a rainfed vineyard located in the Ebro River Basin (NE Spain; 42°02′04′′ N; 0°04′13′′ E) (Figure 1A). This plantation includes four commercial fields (Fábregas Cellar, a winery with Certificate of Origin: Somontano) that are cultivated with the Spanish variety Grenache ('Garnacha Tinta'; *Vitis vinifera* L. cv. Grenache). This site is located on a rolling landscape (mean slope gradient of 9.8%) in the lower part of "Los Oncenos" sub-catchment (see Supplementary Material), within the Vero River catchment, near Barbastro town (Huesca province). Elevation ranges from 447 to 468 m a.s.l. (Figure 1B). Three fields include red grapes (planted in 2008; named VY1, VY2, and VY3), and one is devoted to white grapes (planted in 2007; VY4) under rainfed management. The whole vineyard plantation is arranged in 147 straight lines (espalier system). Soil in the grapevine lines (row hereafter) remains between 8 cm and 23 cm, 13 cm on average, raised related to the soil in the inter-row area, due to the tillage practices carried out by the farmer. Therefore, ground elevation is higher in

the rows than in the inter-row areas. More information about this plantation and the surrounding land uses can be found in López-Vicente and Álvarez [37]. The climate is continental Mediterranean, with an average annual precipitation of 627 mm, mainly concentrated in two rainy seasons, in spring (April–June; 31% of the total annual rainfall) and autumn (September–November; 28%), and the average annual potential evapotranspiration is 1043 mm (data source: "Oficina del Regante" of the Regional Government of Aragon). The summer is dry with occasional thunderstorms and snowfall events are scarce in winter. The mean annual temperature was 14.1 °C between 2004 and 2016.

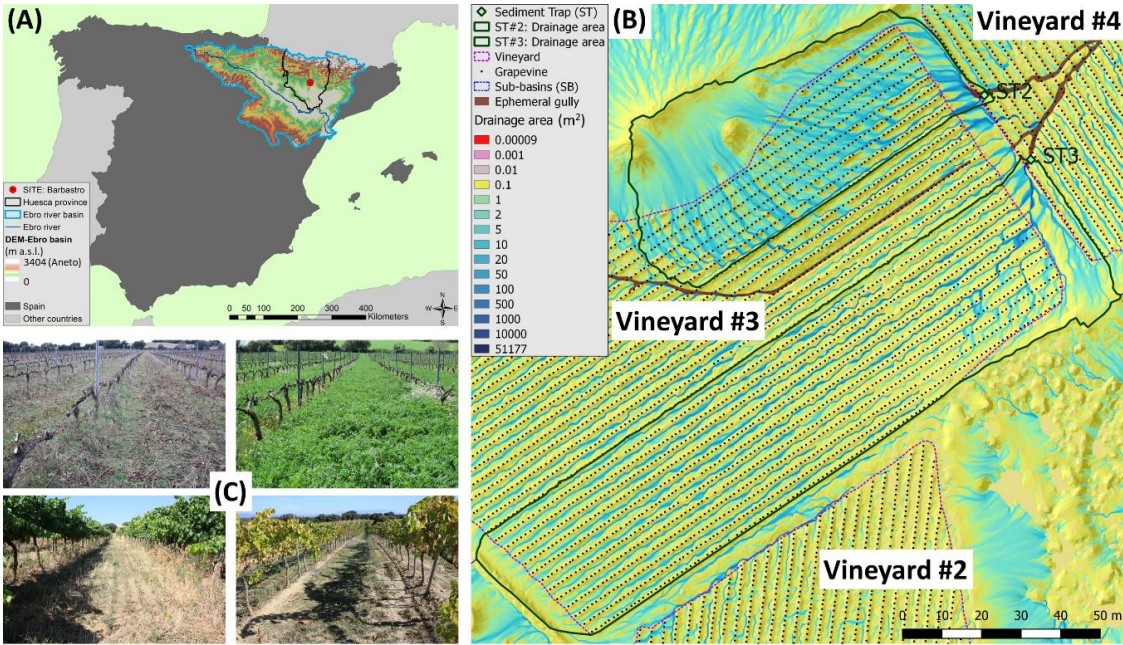

**Figure 1.** (**A**) Map of the overland flow pathways in the part of Los Oncenos sub-catchment where the drainage area of the two sediment traps are located, showing the location of the vines and soil sampling points (**B**). Pictures in January, April, July, and October (**C**).

Four sediment traps were included in the original design of the experiment in Los Oncenos sub-catchment—two in the cereal fields (ST1 and ST4) and two in the vineyards (ST2 and ST3). This research focuses on the vineyards. The two sediment traps (ST2 and ST3) were established in the first inter-row of VY4 down the corridor that separates VY3 and VY4 (Figure 2A). Each ST was located in the course of an ephemeral gully and installed below the soil surface to avoid any disturbance with the tractor traffic (Figure 2B). The drainage areas (DAs) of the ST2 and ST3 were 3286 m² and 6214 m², with a mean slope gradient of 17.0% and 9.2%, respectively (Table 1). In a recent study in the "Los Oncenos" sub-catchment, López-Vicente and Álvarez [38] found that soil roughness and the index of sediment connectivity (IC; water-mediated transfer of soil and sediments within a hydrological unit) are higher in DA-ST2 than in DA-ST3, indicating active processes of sediment delivery. However, DA-ST2 is closer to a convex topography (estimated with the convergence index–CI) that is characteristic of disperse flow at short distance, whereas DA-ST3 has a general concave surface, which indicates concentrated overland flow when neighbor pixels are analyzed. Intense soil erosion processes have launched the development of continuous flow path lines, breaking the topographic sills of the rows in some sections. Soils are Haplic Regosols (calcaric; RGca) in the upper part of DA-ST2 and Luvic Calcisols (CLl) in the lower part of DA-ST2 and in DA-ST3.

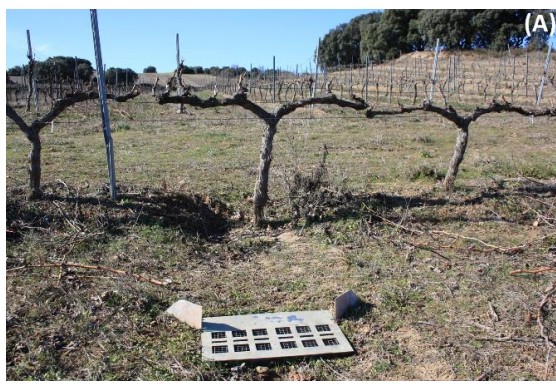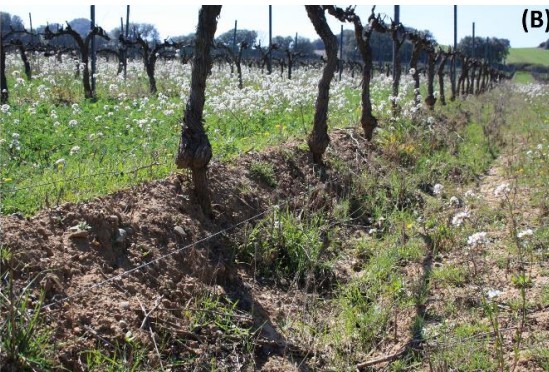

**Figure 2.** Picture of the sediment trap #2 (**A**) and ephemeral gully located in the drainage area of ST3 within VY3 (**B**).

**Table 1.** Physiographic characteristics of the upslope drainage area (DA) of the two sediment traps (STs).

| Sub-Basin | A | S (Mean ± S.D.) | Average Bare Soil | SR | CI | IC (Mean; Max.) |
|---|---|---|---|---|---|---|
| ID | m$^2$ | % | % | mm | % | [$-\infty$; $+\infty$] |
| DA-ST2 | 3286 | 17.0 ± 9.5 | 18.4 | 21.5 | −0.143 | −6.394; −4.695 |
| DA-ST3 | 6214 | 9.2 ± 5.0 | 10.6 | 11.6 | −0.668 | −6.821; −4.818 |

A: total area; S: slope gradient; SR: soil roughness; CI: convergence index; IC: index of runoff and sediment connectivity.

The two STs played the role of collectors of runoff and sediment generated during the different rainfall-runoff events. Each trap had two boxes; one box was buried and permanently remained in the field, and the other one was located inside the first box. This allowed an easy measurement of the runoff and sediments collected during most of the rainfall events but not during the high magnitude-low frequency ones as then the buried box should also be emptied. A metal grid with holes (4 × 4 mm) was located on the top of the smaller box to avoid the entrance of animals and rocks. Finally, a metal cap with big holes (7 × 7 cm) was used to close the trap. Each trap was designed to hold a maximum volume of 32.2 L (460 mm length × 200 mm width × 350 mm depth). Both STs were installed in December 2016 and tested in January 2017. The test period lasted 15 months, from February 2017 to April 2018. During this period and after each heavy rainfall event or after several low- or medium-intensity rainfall events, the runoff and sediment samples were collected and associated to the corresponding time-integrated period (TIP). Every TIP covered the period (in days) between the latest and the new field survey. We performed 26 field surveys. After sampling collection, the internal boxes with runoff and sediments were transported to the laboratory and replaced by new boxes. Then, the total runoff with sediments of each trap was weighted, and sediment was separated by decantation. The wet sediment was dried in an oven at 60 °C for 96 h to ensure a complete dry out of the samples, and the dry sediment was weighed. We selected 60 °C temperature to avoid destroying the organic matter during the drying process. The total runoff (Q; L TIP$^{-1}$) and sediment yield (SY; g TIP$^{-1}$) were calculated for each ST and TIP and presented in a previous study [7] (Table 2). During the test period, values of rainfall depth were obtained from two weather stations, called "Barbastro-Oficina del Regante" (managed by the Regional Government of Aragon; records every 30 min) and "Barbastro-CHEbro" (managed by the Ebro River Basin water authorities; records every 15 min), that are located 4.2 and 5.2 Km eastern from the study area, respectively. Ben-Salem et al. [7] observed minor differences between the records of the two stations and then generated a synthetic weather station. First, the temporal resolution of both records was equalized by calculating the values of precipitation every 30 min. Then, the mean value of precipitation was calculated and assigned this value to the synthetic weather station. Using aggregated values every 30 min, the number of erosive rainfall events and their corresponding rainfall depth (R$_e$; mm), intensity (I$_{30e}$; mm h$^{-1}$), and erosivity (EI$_{30e}$; MJ mm ha$^{-1}$ h$^{-1}$) were calculated.

**Table 2.** Accumulated rainfall depth ($\sum R$) and erosivity ($\sum EI_{30}$) calculated with the synthetic weather station, runoff (*Q*) and sediment (SY) yields, and organic matter content measured in the sediment ($OM_{Sed}$) collected in the two sediment traps (ST) at each time-integrated period (TIP) during the 15-month test period.

| TIP | | $\sum R$ | $\sum EI_{30}$ | GC | ST | Q | SY | $OM_{Sed}$ | |
|---|---|---|---|---|---|---|---|---|---|
| Date | # | mm | MJ mm ha$^{-1}$ h$^{-1}$ TIP$^{-1}$ | Level | # | L TIP$^{-1}$ | g TIP$^{-1}$ | % | g ha$^{-1}$ TIP$^{-1}$ |
| 07/02/2017 | 1 | 24.3 | 10.3 | High | ST2 ST3 | MF 26.488 | ND 16.1 | ND 6.27 | ND 1.6 |
| 16/02/2017 | 2 | 30.1 | 30.4 | High | ST2 ST3 | MF 23.824 | ND 39.8 | ND 3.22 | ND 2.1 |
| 08/03/2017 | 3 | 34.1 | 43.1 | High | ST2 ST3 | MF 3.678 | ND 22.6 | ND 5.07 | ND 1.8 |
| 28/03/2017 | 4 | 70.3 | 68.1 | Very high | ST2 * ST3 * | 30.195 30.010 | 30.9 124.6 | 3.46 3.35 | 3.2 6.7 |
| 17/04/2017 | 5 | 11.7 | 5.1 | Very high | ST2 ST3 | 0 0 | 0 0 | ND ND | ND ND |
| 04/05/2017 | 6 | 23.3 | 29.4 | Very high | ST2 ST3 | 0.720 0 | 3.4 0 | ND † ND | ND † ND |
| 17/05/2017 | 7 | 31,9 | 68.8 | Very high | ST2 ST3 | 0 0.930 | 0 2.9 | ND ND † | ND ND † |
| 06/06/2017 | 8 | 39.9 | 27.9 | Low | ST2 ST3 | 1.500 4.787 | 2.3 7.1 | ND † ND † | ND † ND † |
| 16/06/2017 | 9 | 2.8 | 0.8 | Low | ST2 ST3 | 0 0 | 0 0 | ND ND | ND ND |
| 27/06/2017 | 10 | 19,4 | 47.6 | Low | ST2 ST3 | MF 21.128 | ND 38.9 | ND ND † | ND ND † |
| 10/07/2017 | 11 | 10.8 | 8.0 | Low | ST2 ST3 | 0.383 0.462 | 1.7 1.4 | ND † ND † | ND † ND † |
| 29/08/2017 | 12 | 7.6 | 2.4 | Very low | ST2 ST3 | 0 0 | 0 0 | ND ND | ND ND |
| 19/09/2017 | 13 | 16.7 | 9.0 | Very low | ST2 ST3 | 0.645 0.206 | 27.8 15.2 | 5.48 5.30 | 4.6 1.3 |
| 26/09/2017 | 14 | 18.1 | 50.7 | Very low | ST2 ** ST3 | 9.942 7.071 | 32,825.8 1821.6 | 0.60 3.48 | 599.4 102.0 |
| 17/10/2017 | 15 | 1.3 | 0.1 | Low | ST2 ST3 | 0 0 | 0 0 | ND ND | ND ND |
| 25/10/2017 | 16 | 42.9 | 107.9 | Low | ST2 ** ST3 ** | 8.546 26.513 | 41,260.2 2778.4 | 0.72 4.11 | 904.1 183.8 |
| 17/11/2017 | 17 | 8.0 | 15.3 | Medium | ST2 ST3 | 23.156 8.252 | 281.4 911.6 | 5.34 4.40 | 45.7 64.7 |
| 20/12/2017 | 18 | 13.8 | 2.9 | Medium | ST2 ST3 | 1.012 0.276 | 2.2 0.4 | ND † ND † | ND † ND † |
| 18/01/2018 | 19 | 28.6 | 12.1 | High | ST2 ST3 | 28.396 24.612 | 60.5 38.1 | 6.21 4.63 | 11.4 2.8 |
| 12/02/2018 | 20 | 42.4 | 11.5 | High | ST2 ST3 * | 2.287 28.766 | 0.8 44.6 | ND † ND † | ND † ND † |
| 19/02/2018 | 21 | 9.5 | 5.0 | High | ST2 ST3 | 0 0 | 0 0 | ND ND | ND ND |

**Table 2.** *Cont.*

| TIP | | $\sum R$ | $\sum EI_{30}$ | GC | ST | Q | SY | | OM$_{Sed}$ |
|---|---|---|---|---|---|---|---|---|---|
| Date | # | mm | MJ mm ha$^{-1}$ h$^{-1}$ TIP$^{-1}$ | Level | # | L TIP$^{-1}$ | g TIP$^{-1}$ | % | g ha$^{-1}$ TIP$^{-1}$ |
| 07/03/2018 | 22 | 45.4 | 18.1 | High | ST2 * | 30.711 | 13.3 | 4.25 | 1.7 |
| | | | | | ST3 * | 30.237 | 18.8 | 5.45 | 1.6 |
| 19/03/2018 | 23 | 23.3 | 16.8 | Very high | ST2 | 28.466 | 84.5 | 5.45 | 14.0 |
| | | | | | ST3 | 28.380 | 40.7 | 6.72 | 4.4 |
| 05/04/2018 | 24 | 17.7 | 5.6 | Very high | ST2 | 0.861 | 4.1 | ND † | ND † |
| | | | | | ST3 * | 28.900 | 42.0 | 4.50 | 3.0 |
| 18/04/2018 | 25 | 93.8 | 101.1 | Very high | ST2 * | 28.834 | 59.2 | 5.60 | 10.1 |
| | | | | | ST3 * | 29.034 | 30.2 | 5.51 | 2.7 |
| 30/04/2018 | 26 | 22.4 | 37.6 | Very high | ST2 * | 29.023 | 94.3 | 5.64 | 16.2 |
| | | | | | ST3 * | 27.702 | 572.7 | 7.01 | 64.6 |

GC: ground cover; MF: malfunctioning; *: ST completely full of runoff; **: ST completely full of sediments; †: Insufficient amount of sediment to perform the chemical analysis.

Regarding the vegetation cover, the soil surface of the DA-ST2 had less vegetation with an average percentage of bare soil of 18.4%, whereas the DA-ST3 only had 10.6% of the soil surface without vegetation [7]. However, significant changes in the soil surface cover were observed over the twelve months of the year due to the phenology of the grapevines, the cover crop (CC), the spontaneous vegetation and the tillage practices (e.g., grape harvest and mowing pass) (Figure 1C). The inter-row areas of the vineyards are managed with a mixture of plant species—(i) spontaneous vegetation in VY4 and (ii) plantation of common sainfoin (*Onobrychis viciifolia* Scop., 1772) with spontaneous vegetation in VY1, VY2, and VY3. Spontaneous vegetation also covers the soil in the corridors between the four vineyards. This ground cover was seeded for the first time in early 2016. The maintenance of the CCs includes one mowing pass in spring to avoid water and nutrient competition between the CCs and the vines. In this study, the mowing was done in the third week of May 2017 (later than usual) and in the first week of May 2018. Most pruning remains on the same place after the mowing, so the soil cover factor (percentage of the soil surface covered with vegetation) keeps high all over the year. Herbicide was applied under the vines along the row to avoid the growth of weeds. Grapes were harvested in September.

### 2.2. Organic Matter Content in the Soil and Sediment Samples

In a previous study, López-Vicente and Álvarez [37] collected 144 soil samples in 48 sampling points in the four vineyards and estimated the bulk density (BD), rock content (RF; fragments with a minimum diameter higher than 2 mm), and texture of the soil (content of clay-, silt- and sand-sized particles) at field scape. In this study, the organic matter (OM) content of the soil samples included in the DAs of the two STs was measured. Besides, we generated the corresponding maps of all these physical and chemical properties by means of spatial interpolation. We chose the spline interpolation tool (type: Tension; weight: 100; # points: 1) available in *ArcGIS© 10.5* (ESRI, USA). All maps were generated at 0.2 × 0.2 m of cell size. In the previous cited studies, we observed clear differences in the soil properties between the rows and inter-row areas. Therefore, we decided to use a two-step process: (I) three interpolation maps were generated using the solely points of the different compartments (1) samples from the inter-row areas and corridors; 2) samples from the rows; and 3) samples from the natural vegetation); and (II) overlap the three interpolated maps. Then, the values of the different parameters in the DA-ST2 and DA-ST3 were obtained from the resulting maps.

After removing the coarse fragments (mean diameter higher than 2 mm) of the soil and sediment samples, the fine fractions were analyzed in the certified laboratory "Centro Tecnológico Agropecuario Cinco Villas S.L." Values were presented as weight percentage. During the 26 TIPs, we collected sediment samples in 16 and 20 TIPs in ST2 and ST3, respectively. However, the amount of fine sediment (after removing water, small rocks, and the small vegetation remnants) was very low (SY < 5 g) in some

of them, and the OM content was measured in 10 and 14 samples from the ST2 and ST3, respectively (Table 2). Few differences among the dates of the measured samples appeared due to the different hydrological response of the soils in the DA of the two STs.

*2.3. Organic Matter Enrichment Ratio and Statistical Analysis*

The sediment/soil OM enrichment ratios ($ER_{OM}$) were calculated in the two sediment traps for each TIP,

$$ER_{OM} = \frac{OM_{Sed}}{\overline{OM_{Soil}}} \tag{1}$$

where $OM_{Sed}$ (%) is the concentration of OM in the sediment and $\overline{OM_{Soil}}$ (%) is the average concentration of OM in the soil of the drainage area of each sediment trap.

Then, the mean and total values of OM loss were estimated at annual and test-period scales. The statistical differences of the SY and $ER_{OM}$ between the two STs were analyzed over the test period by means of the analysis of variance (ANOVA; one-way) with the Shapiro¬–Wilk normality test at *p*-value < 0.05.

## 3. Results and Discussion

*3.1. Organic Matter Content in the Sediment Samples*

During the 15-month test period, the accumulated rainfall depth and erosivity was 690.1 mm and 735.3 MJ mm ha$^{-1}$ h$^{-1}$ T$^{-1}$, respectively, which means 552.1 mm yr$^{-1}$ and 588.2 MJ mm ha$^{-1}$ h$^{-1}$ yr$^{-1}$. These values of precipitation were ca. 12% lower than the average rainfall in the region, and higher losses of soil and carbon may be expected in other years. At TIP scale, the concentration of OM in the sediment (grams of OM per grams of sediment) collected in the ST3 ($\overline{OM_{Sed}} = 4.93\% \pm 1.20$) was slightly higher than the concentration of OM in the sediment obtained in the ST2 ($\overline{OM_{Sed}} = 4.28\% \pm 2.06$), but the differences were not significant (*p*-value = 0.336) (Table 2). In the ST2, the OM content ranged between 0.60% and 6.21%, whereas in the ST3, the OM content ranged between 3.22% and 7.01% (Figure 3A). In the collected sediment, the concentration of OM was higher in the ST2 than in the ST3 in five TIPs, and in five TIPs, the concentration of OM was higher in the ST3 than in the ST2. During the five months (from June to October) with the lowest ground vegetation cover (GC) in the inter-row areas, because the CC was mowed in May, the concentration of OM was the lowest in the ST2 ($\overline{OM_{Sed}} = 2.27\%$) and ST3 ($\overline{OM_{Sed}} = 4.30\%$). However, during the four months with moderate GC (November–February) and the three months with maximum GC (March–May), the concentration of OM was clearly higher in the ST2 ($\overline{OM_{Sed}} = 5.78\%$ and 4.88% and ST3 ($\overline{OM_{Sed}} = 4.63\%$ and 5.37%. The differences between the OM densities in the periods with low and moderate-high GC were statistically significant (*p*-value = 0.012), showing the different type of soil erosion that happened over the year. As expected, low values of SY were observed during the months with moderate and high GC in the ST2 ($\overline{SY} = 69$ and 32 g TIP$^{-1}$) and ST3 ($\overline{SY} = 150$ and 85 g TIP$^{-1}$), whereas very high values of SY appeared during the months with the lowest GC in the ST2 ($\overline{SY} = 8235$ g TIP$^{-1}$) and ST3 ($\overline{SY} = 518$ g TIP$^{-1}$). Moreover, the differences of SY between the periods with low and moderate–high GC were statistically significant (*p*-value = 0.005). These results totally agree with those observed in vineyards [39] and olive groves [40] managed with temporary CC, where the positive effect of CC on soil erosion control declined after mowing or application of herbicides. On the other hand, low-intensity erosive events detach fine soil particles, which gather most SOM [41], whereas high-intensity erosive events detach and transport sediment particles of all sizes [42], including sand and rocks that are poor in SOM. The combined effect of preferential soil particle size remobilized during the distinct rainfall events and the temporal changes in ground cover explains the temporal pattern of OM concentration in the sediment. These temporal patterns were even more notably in the ST2 than in the ST3, suggesting that higher values of GC reduce the temporal variability of soil and OM loss, and agree with the results of higher runoff stability observed by López-Vicente et al. [22] in

olive groves with CC compared with olive groves under CT. On the other hand, the spatial pattern of soil redistribution strongly depends on the magnitude and intensity of the rainfall event, and clear changes can be found within the same site [43]. Therefore, collected runoff and sediment during the low intensity events may correspond to the drainage located near the ST. Most likely, only during the heaviest storm events, eroded particles from any part of the drainage area reached the ST. Further research should focus on identifying the travel distance of detached soil.

Regarding runoff (Q; L $TIP^{-1}$) and sediment yield (SY; g $TIP^{-1}$), we observed some positive links between Q and the concentration of OM in the sediment (Pearson's $r = 0.324$) (Figure 4A), and a negative and stronger relationship between SY and the concentration of OM in the sediment ($r = -0.769$) (Figure 4B). The relationship between the accumulated rainfall erosivity ($\Sigma EI_{30}$) and the concentration of OM was also negative ($r = -0.415$), mirroring the pattern of SY vs. OM, so the highest OM concentrations appeared in the events with low and medium values of $EI_{30}$. The highest concentrations of OM, between 5% and 7%, appeared with the lowest values of SY, whereas the OM concentration diminished below 4% with moderate-high values of SY and decreased below 1% in the two TIPs with the highest amounts of eroded soil (registered in September and October in the ST2). This relationship was stronger in the ST2 ($r = -0.916$) than in the ST3 ($r = -0.332$). These results may be explained by the non-linear relationship that exists between runoff and sediment yield observed by different authors in distinct sites [44]. Besides, our results are in accordance with the previous analysis of the temporal pattern of OM loss and agree with the assessment of the long-term impact of reduced tillage on soil and nutrient losses performed by Klik and Rosner [45] in Lower Austria. These authors found that conservation tillage favored lower soil loss rates, compared with conventional tillage, even though nutrients (N and P) concentrations in runoff were greater in the conservation tillage fields than in the CT fields and sediments were clearly enriched in N and P.

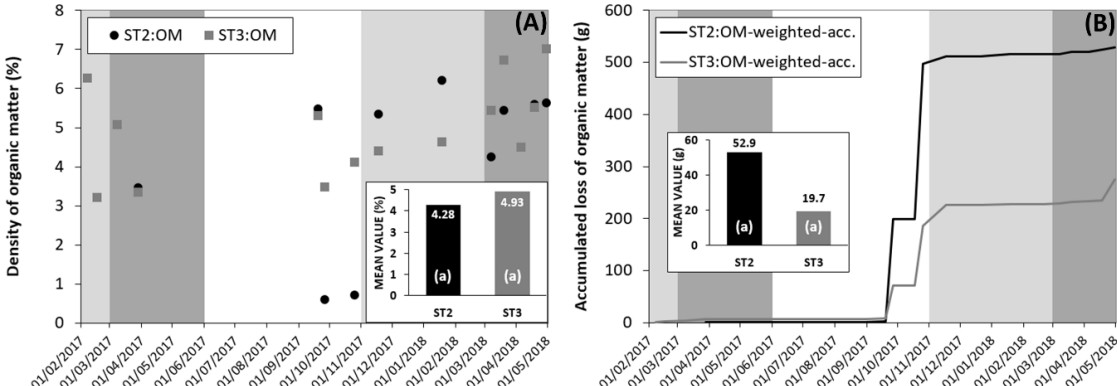

**Figure 3.** Concentration or density of organic matter (OM) in the sediment (in %) during the test period (**A**). Cumulative total loss of OM (in grams) in the ST2 and ST3 during the test period (**B**). The white, light grey, and grey areas correspond to the periods with low, medium, and high ground vegetation cover, respectively. The same lowercase letters in the inset figures indicate that no significant difference appeared between the ST2 and ST3 at 0.05 level.

Accounting the total weight of the sediments (total loss of OM), our results clearly changed due to the higher amount of soil loss from the CA-ST2. The total loss of OM during the test period was of 529.2 g in the ST2 (423.4 g $yr^{-1}$) of 275.3 g in the ST3 (220.3 g $yr^{-1}$) (Figure 3B). Despite the total amount of OM loss in the ST2 being 1.9 times higher than in the ST3, the differences between the two STs were not significant ($p = 0.277$). Based on the extent of the drainage area of the two STs, the average annual rate of OM loss was of 1.29 and 0.35 $kg_{OM}$ $ha^{-1}$ $yr^{-1}$ in the ST2 and ST3. Although the average annual rate of OM loss in the ST2 was 3.6 times higher than in the ST3, the differences between the two STs during the test period were not significant ($p = 0.149$). These observed rates were similar to the losses of dissolved organic carbon (ca. 7.8 $kg_{DOC}$ $ha^{-1}$ $yr^{-1}$) and lower than the losses of total organic

carbon (ca. 101 $kg_{TOC}$ $ha^{-1}$ $yr^{-1}$) measured by Gómez et al. [46] in Spanish olive orchard plots with CC. Besides, these authors also reported clear benefits on reducing the loss of carbon in those plots with CC compared with the plots under CT. In vineyards located in Central Spain, Ruiz-Colmenero et al. [24] also found that the concentration of SOC in sediments was greater in the cover crops plots than in the CT plots, whilst the mass-corrected loss of SOC was greater under CT (ca. 60 $kg_{OM}$ $ha^{-1}$ $yr^{-1}$) than under CC (ca. 20 $kg_{OM}$ $ha^{-1}$ $yr^{-1}$).

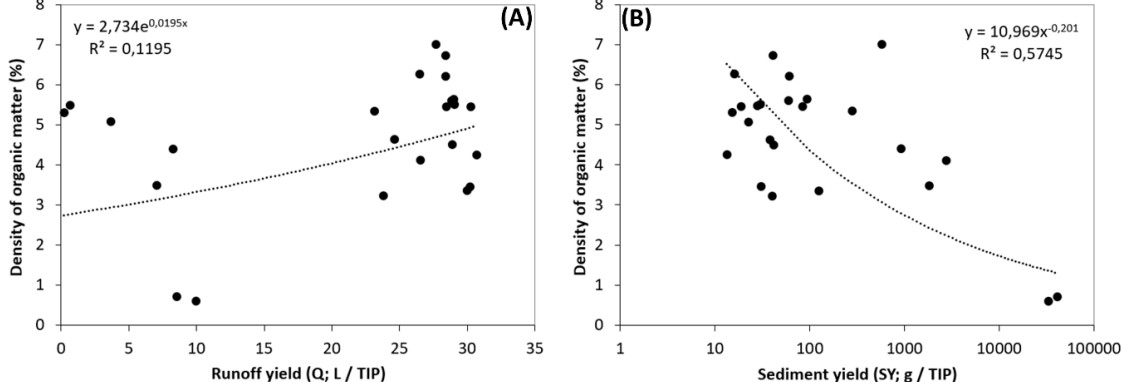

**Figure 4.** Relationship between the concentration or density of OM in the sediment and the amount of runoff (Q) (**A**) and sediment yield (SY) (**B**) per time-integrated period (TIP).

### 3.2. Soil Physical and Chemical Properties

The interpolated maps allowed estimating the spatially distributed values of the soil physical and chemical properties (Figure 5). Regarding the physical parameters, the mean values of bulk density, effective volume of the soil (volume occupied by the fine fraction, mean diameter lower than 2 mm), silt and clay contents were 1.22, 1.08, 1.46 and 1.41 times higher in the CA-ST3 than in the CA-ST2 (Table 3). Rock fragments and sand content were 2.71 and 1.37 times higher in the CA-ST2 than in the CA-ST3. The dominant soil texture is sandy loam in the CA-ST2 and loam in the CA-ST3. The lower content of silt and clay found in the DA-ST2 can be associated with the higher soil loss rates observed in this area because continuous water soil erosion tends to reduce the amount of fine soil particles in the topsoil layer [42].

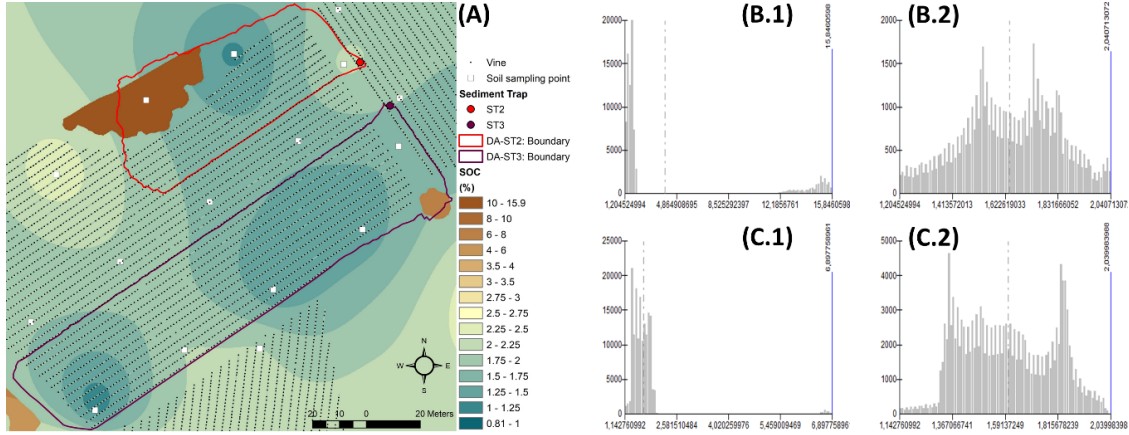

**Figure 5.** Map of the soil organic matter content (**A**) and corresponding histograms of the DA-ST2 (**B**) and DA-ST3 (**C**) in the total drainage area (**B.1** and **C.1**) and within the vineyard (**B.2** and **C.2**).

**Table 3.** Physical and chemical soil parameters in the drainage area of the two sediment traps.

| Sub-basin | BD | RF | Vol$_{eff}$ | Clay | Silt | Sand | Texture | SOM |
|---|---|---|---|---|---|---|---|---|
| ID | gr·cm$^{-3}$ | % Weight | % Vol. | % | % | % | Type | % |
| DA-ST2: all area | 1.190 | 34.2 | 86.19 | 6.34 | 31.22 | 62.44 | Sandy loam | 4.02 ± 5.10 |
| DA-ST2: vineyard | 1.286 | 29.0 | 86.76 | 6.95 | 32.99 | 60.06 | Sandy loam | 1.64 ± 0.19 |
| DA-ST3: all area | 1.450 | 12.6 | 92.68 | 8.96 | 45.54 | 45.50 | Loam | 1.66 ± 0.60 |
| DA-ST3: vineyard | 1.448 | 12.5 | 92.74 | 9.01 | 45.64 | 45.35 | Loam | 1.60 ± 0.19 |

BD: bulk density; RF: rock fragments; Vol$_{eff}$: effective volume; SOM: Soil organic matter content.

The SOM content in the DA-ST2 ($\overline{OM_{Soil}} = 4.02$) was higher than in the DA-ST3 ($\overline{OM_{Soil}} = 1.66$) due to the presence of a small forest area in the upper part of the drainage area (Table 3). The soil in this patch of Valencian oaks (*Quercus faginea*; a species of oak native to the western Mediterranean region in the Iberian Peninsula) have high values of SOM (between 11.2% and 15.8%). When the spatial analysis was limited to the area occupied by the vineyard, the SOM content was almost the same in the DA-ST2-vineyard ($\overline{OM_{Soil}} = 1.64$) and DA-ST3-vineyard ($\overline{OM_{Soil}} = 1.60$). These results did not reflect higher values of SOM in the DA with higher ground cover and partially disagree with those obtained by Novara et al. [8], who found that, after five years of cover crop soil management in Italian vineyards, there were higher SOC contents in the CC fields compared to the CT fields. That fact could be explained by the recent short duration of the CC in the study area, lower than one year at the beginning of the field surveys and around two years at the end of the study. Therefore, more marked differences are expected to be found in the next years when the CC implementation reach a longer duration.

*3.3. OM Enrichment Ratio, Cover Crops, and SDGs*

When only the drainage area within the vineyard of each ST was considered, the ER$_{OM}$ was similar between the two STs—2.61 and 3.07 in ST2 and ST3, respectively (Figure 6A). However, when the total drainage area of the two STs were included in the analysis, clear differences appeared between the two STs: 1.06 and 2.97 in ST2 and ST3, respectively. At TIP scale and accounting the drainage area within the vineyard, the ER$_{OM}$ ranged between 0.37 and 3.79 in ST2 and between 2.01 and 4.37 in ST3 (Figure 6B). Ruiz-Colmenero et al. [36] found similar ER$_{OM}$ in vineyards located in Central Spain, with ratios ranging from 1.4 to 2. The clear temporal changes in the SY and the concentration of OM$_{Sed}$ and ER$_{OM}$ over the test period suggest that the management of the CC as a permanent cover instead of the current use as a temporary cover could help minimize the loss of soil and OM in the intense rainfall events that commonly happen in autumn in the study area. Schütte et al. [47] recommended this practice after evaluating soil loss at two different European wine regions (one in Spain and another in Austria).

Our research at the Somontano Certified Region in Aragón, where the rainfed vineyard in rolling landscapes produces high soil losses, demonstrates that there are solutions to achieve a sustainable management. The use of a cover crop of common sainfoin (*Onobrychis viciifolia* Scop., 1772) in the inter-row area of the vineyards contributed to reduce both the soil and OM losses (from 1.29 to 0.35 kg$_{OM}$ ha$^{-1}$ yr$^{-1}$). This reduction in 3.6 times was also found when the soil losses in other crops were measured. Cerdà et al. [48] measured very low soil erosion rates (0.12 Mg ha$^{-1}$ h$^{-1}$) in citrus orchards when covered by weeds or catch crops. However, similar rainfall simulation experiments contributed 15.7 Mg ha$^{-1}$ h$^{-1}$ in a clementine plantation managed with herbicides, where no plant cover was present. The key role that vegetation plays in agriculture land has been also assessed by Malik et al. [49] with the use of *Lolium multiflorum* L., *Festuca arundinacea* L., *Trifolium incarnatum* L., and *Lespedeza cuneata* (Dumont) G. Don. along 585 days, measuring a reduction in 64%, 61%, 51%, and 37% in soil erosion, respectively.

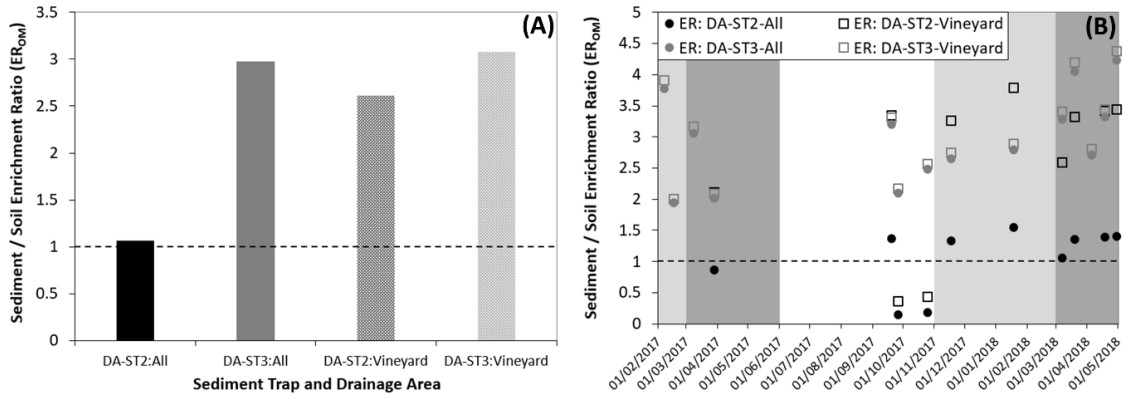

**Figure 6.** Mean sediment/soil organic matter enrichment ratio ($ER_{OM}$) in the ST2 and ST3 considering the whole drainage area or the drainage area within the vineyard (**A**). Evolution of the $ER_{OM}$ at TIP scale during the text period (**B**).

The positive impact of CC in the control of soil erosion is a consequence of two different processes. The first one is the protective cover of the plant that reduce or avoid the raindrop impact and the surface wash is also reduced due to the lower runoff velocity as a consequence of the stems of the plants. The rainfall interception of the plants also reduces the effective rainfall that reach the soil. This is the main impact of the straw cover that acts as a mulch and is being applied as a quick strategy to control soil loss [50]. However, the vegetation cover is also relevant to improve soil quality. Vegetation is a key component to reduce soil losses, as has been demonstrated here, but the plant cover and the root system also induce an increase in the soil biota, organic matter, aggregate stability, and soil porosity. Van Hall et al. [51] found an increase in soil organic carbon, total nitrogen, and aggregate stability and a decrease in soil bulk density and pH along a vegetation succession. This was a long-term impact of 50 years. At the seasonal scale, Wu et al. [52] measured an increase in the soil quality due to the soil microbial biomass and community structure enhancement as a consequence of the Kudzu vine cover in the Jinsha River Valley in southwest China.

The impact of the CC researched at the Somontano wine production area affect the soil clearly reducing the losses of soil matter, and thus, of organic carbon. This impact will accumulate yearly. Furthermore, there will be an improvement of the soil quality that will result in the reduction of the soil losses to low values, as the long-term measurements demonstrate. The impact of the use of mulches or cover crops induce a reduction in soil losses in the short term, and after a few years, the soil is improved due to the reduction of soil and carbon (and water, seeds, biota, etc.) losses. That is what Derpsch et al. [53] found between 1977 and 1984 in Paraná (Brazil), which is related to the findings of Mazzoncini et al. [54] regarding the link to the improvement in soil organic cover and total nitrogen content or the findings of Sainju et al. [55] in sandy loam soil in Georgia, USA.

Our findings and the literature review confirm that cover crops are an option to change unsustainable agriculture management into a management that can contribute to achieve the SDGs of the United Nations and the Land Degradation Neutrality challenge [5]. This should be related to the use of more environmentally friendly management strategies in agriculture such as organic farming and regenerative agriculture [56,57]. This is especially relevant for vineyards due to the high erosion rates measured in different parts of the world where rainfall simulator experiments [58], plots under natural rainfall [59], modelling [60], and long-term measurement strategies such as botanical bench-marks [61] and the improved stock-unearthing method (ISUM) [62] demonstrate that the current situation of the vineyards is not sustainable.

## 4. Conclusions

To reach the SDGs of the United Nations, transitional changes in the farm management are needed to achieve sustainable production of food and reduce the impact generated by crop management that

threatens soil health. Soil organic matter is a key component of the soil system that needs to be enhanced to reduce the impact of the global warming and reach land degradation neutrality. Our research demonstrates that the use of cover crops contributes to reduce soil organic matter losses. Despite similar concentrations of organic matter in the soil and sediments that appeared in distinct areas within the vineyard, the annual rates of sediment yield and soil organic matter losses were clearly higher in plots with lower plant cover. Over the experimental period, the concentration of organic matter in the sediment showed a negative relationship with the net soil loss. The observed significant differences between the content of organic matter in the sediment in the months with low and moderate-to-high ground cover suggested that permanent cover crops should be considered to extend the benefits of the ground cover. Overall, cover crops are an efficient strategy to control unsustainable soil and organic matter losses measured in vineyards, and this is a key contribution to achieve the SDGs of the United Nations before 2030.

**Supplementary Materials:** The following Google Earth file is available online: http://www.mdpi.com/2073-445X/9/7/230/s1.

**Author Contributions:** All authors made substantial contributions to this work. In particular, M.L.-V. conceived the study and participated in all tasks, E.C.-S. participated in the data analysis and mapping, S.Á. participated in the field surveys and data processing, and A.C. participated in the interpretation of the data and results and the writing of the manuscript. All authors have read and agreed to the published version of the manuscript.

**Funding:** This research was funded by the project "Environmental and economic impact of soil loss (soil erosion footprint) in agro-ecosystems of the Ebro river basin: numerical modelling and scenario analysis (EroCostModel) (CGL2014-54877-JIN)" of the Spanish Ministry of Economy and Competitiveness.

**Acknowledgments:** We thank Gonzalo Alcalde Fábregas (Fábregas Cellar, D.O. Somontano) for permitting the use of the vineyards where this research was done. Three reviewers contributed with constructive criticism to this paper. S.D. Keesstra kindly improved the last draft manuscript.

**Conflicts of Interest:** The authors declare no conflict of interest. The funders had no role in the design of the study; in the collection, analyses, or interpretation of data; in the writing of the manuscript, or in the decision to publish the results.

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
