# Peer review of "Effectiveness of Cover Crops to Reduce Loss of Soil Organic Matter in a Rainfed Vineyard"

_land, doi:10.3390/land9070230_

Round 1

Reviewer 1 Report

Dear Authors,

I was pleased to read your manuscript. I have only a few questions and comments.

  1. In my opinion, the description of methods or possibly discussions should mention the origin of the sediment collected for sediment traps. Does it really reflect the quantity and characteristics of the soil eroded from the whole area of the vineyard or does this soil come only from erosion in ephemeral gully? After more rainfall, were observations made about the rills arising within the vineyard? The studied vineyards, especially No. 2, are of considerable size and the question arises to what extent the collected data are representative of its entire area. It is imporatant from the point of view of significance of the results of the study.
  2. Introduction is extensive and contains numerous references to previous tests, including those described above. Therefore, it would be advisable to emphasize the novelty of the study more strongly.
  3. To what extent were the climatic conditions (precipitation) occurring during the study period representative for the climate of this area? The amount of precipitation affects the intensity of erosion.
  4. The legend to Figure 1 is not entirely clear to me. What does the term "drainage area" mean? The larger vineyard has an area of 6000 m2, while the figure shows the value of 50,000 m2. It is not clear. It is also difficult to see red and pink surfaces that appear in the legend to the figure.
  5. Table 1. Please explain what the terms "CI" and "IC" means. How they are calculated and what they really show.

Author Response

Reviewer #1

Dear Authors,

Comment 1: I was pleased to read your manuscript. I have only a few questions and comments.

Reply to R1.1: Thank you very much for your words and dedicated time to revise the manuscript and provide useful comments.

Comment 2: In my opinion, the description of methods or possibly discussions should mention the origin of the sediment collected for sediment traps. Does it really reflect the quantity and characteristics of the soil eroded from the whole area of the vineyard or does this soil come only from erosion in ephemeral gully? After more rainfall, were observations made about the rills arising within the vineyard? The studied vineyards, especially No. 2, are of considerable size and the question arises to what extent the collected data are representative of its entire area. It is important from the point of view of significance of the results of the study.

Reply to R1.2: We appreciate this comment because travel distance is a non-solved question in soil erosion studies. We have added the following text –and a new reference– in the results discussion: “On the other hand, the spatial pattern of soil redistribution strongly depends on the magnitude and intensity of the rainfall event and clear changes can be found within the same site [43]. Therefore, collected runoff and sediment during the low intensity events may correspond to the drainage are located near the ST. Most likely, only during the heaviest storm events, eroded particles from any part of the drainage area reached the ST. Further research should be focus on identifying the travel distance of detached soil.” The new reference is: “Severe soil erosion during a 3-day exceptional rainfall event: Combining modelling and field data for a fallow cereal field. Hydrological Processes 2015, 29(10), 2358–2372. https://doi.org/10.1002/hyp.10370”.

Comment 3: Introduction is extensive and contains numerous references to previous tests, including those described above. Therefore, it would be advisable to emphasize the novelty of the study more strongly.

Reply to R1.3: Thank you for this comment. We have added the following sentences in the latest paragraph of the revised version: “Most previous studies have dealt with this issue at plot or field scale. The novel aspect of this study is the assessment on how the internal changes within the same field influence the magnitude of the benefits of CC on soil and carbon loss reduction.”

Comment 4: To what extent were the climatic conditions (precipitation) occurring during the study period representative for the climate of this area? The amount of precipitation affects the intensity of erosion.

Reply to R1.4: We totally agree with this comment. We have clarified this aspect in the revised version, as follows: “During the 15-month test period, the accumulated rainfall depth and erosivity was 690.1 mm and 735.3 MJ mm ha–1 h–1 T–1, respectively, which means 552.1 mm yr–1 and 588.2 MJ mm ha–1 h–1 yr–1. These values of precipitation were ca. 12% lower than the average rainfall in the region, and thus, higher losses of soil and carbon could be expected in other years.”

Comment 5: The legend to Figure 1 is not entirely clear to me. What does the term "drainage area" mean? The larger vineyard has an area of 6000 m2, while the figure shows the value of 50,000 m2. It is not clear. It is also difficult to see red and pink surfaces that appear in the legend to the figure.

Reply to R1.5: Thank you very much for this remark. The legend of the drainage area corresponds to Los Oncenos sub-catchment. We have added this information in the revised figure caption to avoid any misunderstanding. The red and pink pixels are difficult to distinguish due to the pixel size (0.04 m2) and the scale of the map. These pixels only appear near the divides and are not frequent.

Comment 6: Table 1. Please explain what the terms "CI" and "IC" means. How they are calculated and what they really show.

Reply to R1.6: Thank you for this comment. We have improved the text as follows: “… the index of sediment connectivity (IC; water-mediated transfer of soil and sediments within a hydrological unit) are higher in DA-ST2 than in DA-ST3, indicating active processes of sediment delivery. However, DA-ST2 is closer to a convex topography (estimated with the convergence index – CI) that is characteristic of disperse flow at short distance, whereas DA-ST3 has a general concave surface, which indicates concentrated overland flow when neighbour pixels are analysed.”

Reviewer 2 Report

Review on land-864647 "Effectiveness of cover crops to reduce the loss of soil organic matter in a rainfed vineyard" by López-Vicente et al.

Overall comments

A field study on water erosion in a Spanish vineyard is presented to demonstrate the effect of cover crops on SOM loss under rainfed conditions.  The fieldwork is well organized and results well presented. However, before the paper can be considered further, a range of writing-related issues have to be dealt with. These include (1) the issue of scientific writing. The text is generally understandable, but there is quite a need to improve clarity. There are primitive language errors that are easy to correct by careful revision. More attention is needed on the writing style. My suggestion is to solicit proofreadings by experienced colleagues before resubmission; (2) the hypothesis lacked support from the literature review. The current introduction is mostly a simple literature review without analysis. As a result, the hypothesis that the authors put forward is logically disconnected from the literature review. My suggestion is to adjust the literature review by paying attention to what is needed but NOT available in the literature. It is also possible that what you found in the literature is not consistent, which can give rise to your hypothesis; and (3) clarifications of some of the methods used. The current description of the TIP, synthetic weather station, and ERom are too simple, lacking the required details. Additional descriptions are needed to make these methods explicit. 

Specific comments

Abstract: Results presented in the abstract do not reflect the influence of CC. What do you mean by "loss of soil in permanent crops"? Perhaps "loss of soil in permanent-crop systems" is more explicit? Please check the language used here: "use to" is not correct. SDG should be literally "Sustainable Development Goals".

L35 response: respond;

L38-43 Break the whole sentence (L38-43) into shorter statements;

L40 regulate: regulating;

L42 transition: transit;

L43-44 Check syntax;

L47 have: has;

L47 explains: explain;

L56 with: to;

L59 runoff coefficient: explain what coefficient measures. Does it mean 7.4% of soil mass was lost or soil erosion occurs on 7.4% of the area?

For the rest of the text, I'll skip the language errors, concentrating more on the paper itself. This doesn't mean that there are no language errors;

L60 shown: showed;

Section 2.1: Two sediment traps were installed, ST2 and ST3. Where is ST1?

L136-137 What do you mean? Are you talking about row spacing?

L142 mm-1: no need. delete;

L168 TIP: What a TIP is and how the association was done? TIP is quite important in the following sections, so a clear definition must be given;

L172 oven at 60 C: for how long?

L177 synthetic weather station was generated: How? Better give a brief description and provide references; 

Figure 1: "Study area". The panels are marked with A, B, C, so the same capital letters should be used in the caption;

L214 "is presented": Do you mean "are measured"? If so, measured using which method?

L233 ERom is a central metric used. An equation of mathematical definition should be given;

L241 density: I think it should be "content", or as you indicated (L242) "concentration". It is not density. Replace across the text;

L245-246 Who is higher than whom?

L247 GC is lowest in summer?

L286 "of": and;

L287 "2.7 times": How did you arrive at this number?

L291 "and lower than": Who is lower? Check syntax!

Caption Figure 3 "different lowercase letters": It is not clear which letters you referred to;

Table 3 footnote: What is the meaning of "effective volume"?

L420 "and not temporary": delete;

Captions of all figures and tables: improve the clarity and readability. Pay attention especially to the description of the subpanels inside a figure.

Author Response

Reviewer #2

Comment 1: A field study on water erosion in a Spanish vineyard is presented to demonstrate the effect of cover crops on SOM loss under rainfed conditions.  The fieldwork is well organized and results well presented. However, before the paper can be considered further, a range of writing-related issues have to be dealt with. These include (1) the issue of scientific writing. The text is generally understandable, but there is quite a need to improve clarity. There are primitive language errors that are easy to correct by careful revision. More attention is needed on the writing style. My suggestion is to solicit proofreadings by experienced colleagues before resubmission;

Reply to R2.1: Thank you for your words, positive evaluation and advice. An experienced colleague has checked the whole text and revised the grammar and style.

Comment 2: (2) the hypothesis lacked support from the literature review. The current introduction is mostly a simple literature review without analysis. As a result, the hypothesis that the authors put forward is logically disconnected from the literature review. My suggestion is to adjust the literature review by paying attention to what is needed but NOT available in the literature. It is also possible that what you found in the literature is not consistent, which can give rise to your hypothesis;

Reply to R2.2: We are partly in agreement with this comment. Introduction is divided into six paragraphs. The first one is directly related to the topic of the Special Issue ‘Land Degradation Neutrality’ where this article was submitted, focus on agricultural soils and soil erosion. The second paragraph deals with soil erosion in permanent crops. The third and fourth paragraphs are about the use of cover crops and its benefits to reduce soil and organic matter losses, with special attention in vineyards and other permanent crops. The fifth paragraph is about the physiographic aspects that should be analysed in further studies to advance on soil erosion and carbon studies. In the sixth paragraph we presented the concept of organic matter enrichment ration, the necessity of the research, the hypothesis, the tasks and the expected contribution. However, we agree with Reviewer #1 that the text can be improved, and thus, we have added the following text in the latest paragraph: “Most previous studies have dealt with this issue at plot or field scale. The novel aspect of this study is the assessment on how the internal changes within the same field influence the magnitude of the benefits of CC on soil and carbon loss reduction.”

Comment 3: (3) clarifications of some of the methods used. The current description of the TIP, synthetic weather station, and ERom are too simple, lacking the required details. Additional descriptions are needed to make these methods explicit.

Reply to R2.3: Thank you for your feedback. We have improved the description and explanation of the synthetic weather station, the time-integrated periods (TIPs) and organic matter enrichment ratio (EROM) in different sections and paragraphs across the revised manuscript.

Comment 4: Abstract: Results presented in the abstract do not reflect the influence of CC. What do you mean by "loss of soil in permanent crops"? Perhaps "loss of soil in permanent-crop systems" is more explicit? Please check the language used here: "use to" is not correct. SDG should be literally "Sustainable Development Goals".

Reply to R2.4: We have changed the text according to your suggestions.

Comment 5: L35 response: respond;

Reply to R2.5: We have made the change.

Comment 6: L38-43 Break the whole sentence (L38-43) into shorter statements;

Reply to R2.6: We have divided the original sentence into two sentences.

Comment 7: L40 regulate: regulating;

Reply to R2.7: We have made the change.

Comment 8: L42 transition: transit;

Reply to R2.8: We have made the change.

Comment 9: L43-44 Check syntax;

Reply to R2.9: We have rewritten and improved the sentence.

Comment 10: L47 have: has;

Reply to R2.10: We have made the change.

Comment 11: L47 explains: explain;

Reply to R2.11: We have made the change.

Comment 12: L56 with: to;

Reply to R2.12: We have made the change.

Comment 13: L59 runoff coefficient: explain what coefficient measures. Does it mean 7.4% of soil mass was lost or soil erosion occurs on 7.4% of the area?

Reply to R2.13: We have added the meaning of runoff coefficient: “ratio between the amount of runoff volume and rainfall depth per unit of area”

Comment 14: For the rest of the text, I'll skip the language errors, concentrating more on the paper itself. This doesn't mean that there are no language errors;

Reply to R2.14: Thank you for your comments regarding language errors. We have carefully revised the text to avoid further errors.

Comment 15: L60 shown: showed;

Reply to R2.15: We have made the change.

Comment 16: Section 2.1: Two sediment traps were installed, ST2 and ST3. Where is ST1?

Reply to R2.16: We have added the complete explanation of the sediment traps in Los Oncenos sub-catchment: “Four sediment traps were included in the original design of the experiment in Los Oncenos sub-catchment: Two in the cereal fields (ST1 and ST4) and two in the vineyards (ST2 and ST3). Due to economic restrictions associated with the cost of the chemical analysis of the soil and sediment samples, research was only focused on the vineyards.”

Comment 17: L136-137 What do you mean? Are you talking about row spacing?

Reply to R2.17: It is not about row spacing, it is about the ground elevation in the rows. To avoid any misunderstanding, we have added a new sentence: “Therefore, ground elevation is higher in the rows than in the inter-row areas.”

Comment 18: L142 mm-1: no need. delete;

Reply to R2.18: We have done this change.

Comment 19: L168 TIP: What a TIP is and how the association was done? TIP is quite important in the following sections, so a clear definition must be given;

Reply to R2.19: We explained the meaning of the TIP and the protocol that we followed to do the 26 field surveys.

Comment 20: L172 oven at 60 C: for how long?

Reply to R2.20: We have added the following text in the revised version: “for 96 hours (4 days) –to ensure a complete dry out of the samples–”.

Comment 21: L177 synthetic weather station was generated: How? Better give a brief description and provide references;

Reply to R2.21: We have added new information about the two weather stations used to generate the synthetic weather station: “During the test period, values of rainfall depth were obtained from two weather stations, called ‘Barbastro-Oficina del Regante’ (managed by the Regional Government of Aragon; records every 30 min) and ‘Barbastro-CHEbro’ (managed by the Ebro River Basin water authorities; records every 15 min), that are located 4.2 and 5.2 Km eastern from the study area, respectively. Ben-Salem et al. [7] observed minor differences between the records of the two stations, and then, generated a synthetic weather station.”

Comment 22: Figure 1: "Study area". The panels are marked with A, B, C, so the same capital letters should be used in the caption;

Reply to R2.22: Thank you for this comment. We have revised and corrected all figure captions.

Comment 23: L214 "is presented": Do you mean "are measured"? If so, measured using which method?

Reply to R2.23: We have made this change. The chemical analysis of the soil and sediment samples is explained in the section “2.3. Organic matter enrichment ratio and statistical analysis”

Comment 24: L233 ERom is a central metric used. An equation of mathematical definition should be given;

Reply to R2.24: We have added the equation and the corresponding explanation: “The sediment/soil OM enrichment ratios (EROM) were calculated in the two sediment traps for each TIP:

                                                                                                                                      (1)

where OMSed (%) is the concentration of OM in the sediment and  (%) is the average concentration of OM in the soil of the drainage area of each sediment trap.

Comment 25: L241 density: I think it should be "content", or as you indicated (L242) "concentration". It is not density. Replace across the text;

Reply to R2.25: The term “density” is accepted and used in studies of organic matter and carbon content in soil and sediment samples. However, we have replaced the term “density” by “concentration” in the text.

Comment 26: L245-246 Who is higher than whom?

Reply to R2.26: We have clarified the meaning of this sentence as follows: “In the collected sediment, the concentration of OM was higher in the ST2 than in the ST3 in five TIPs, and…”

Comment 27: L247 GC is lowest in summer?

Reply to R2.27: Thank you for this comment. We clarified that the lowest GC in summer was only related to the inter-row areas, where the cover crop was seeded, and not in the rows where the vines are located. We have improved the text as follows: “During the five months (from June to October) with the lowest ground vegetation cover (GC) in the inter-row areas –because the CC was mowed in May–, the concentration of OM was the lowest…”

Comment 28: L286 "of": and;

Reply to R2.28: Thank you. We have checked the grammar.

Comment 29: L287 "2.7 times": How did you arrive at this number?

Reply to R2.29: In the previous sentence we included the numbers that we used to calculate this ratio.

Comment 30: L291 "and lower than": Who is lower? Check syntax!

Reply to R2.30: We compared our results with those obtained by Gómez et al. (2017). To avoid any misunderstanding we have modified the text as follows: “These observed rates were similar to the losses of dissolved organic carbon (ca. 7.8 kgDOC ha−1 yr−1) and lower than the losses of total organic carbon…”

Comment 31: Caption Figure 3 "different lowercase letters": It is not clear which letters you referred to;

Reply to R2.31: We appreciate this remark. We have improved the text of the figure caption as follows: “The same lowercase letters in the inset figures indicate that no significant difference appeared between the ST2 and ST3 at 0.05 level.”

Comment 32: Table 3 footnote: What is the meaning of "effective volume"?

Reply to R2.32: We defined "effective volume" in the first paragraph of the section 3.2. Soil physical and chemical properties: The effective volume of the soil is the volume occupied by the fine fraction, mean diameter lower than 2 mm).

Comment 33: L420 "and not temporary": delete;

Reply to R2.33: We have made this change.

Comment 34: Captions of all figures and tables: improve the clarity and readability. Pay attention especially to the description of the subpanels inside a figure.

Reply to R2.34: We have checked all captions.

Reviewer 3 Report

This paper studies the rainfed vineyard in a typical Mediterranean climate region and the effects of cover crops on soil organic matter, which has certain value for land degradation and sustainable management. This research is suitable for publication in Land. However, this paper still needs some improvement. My comments and suggestions are as follows:

1.In the abstract(P3L102-104 and P3L115-117), the authors described that the slope gradient also plays an important role in the effect of cover crops on soil organic matter. However, the relevant factors have not been analyzed in this paper. Please explain why.

2.In the legend of Figure 1a on P6, DEM data should be marked with units in parentheses. In the legend of Figure 1b, the unit of drainage area is wrong. It is recommended to use full name or abbreviation for the scale unit of the drawings in the paper, and unify the case of the marked letters in the drawing and the letters in the drawing title.

3.In L268-271 on P7, the authors found that the OM concentration in the sediment was positively correlated with the runoff yield and negatively correlated with the sediment yield. However, the author only made a data description. Please analyze the cause of this phenomenon combined with the field situation and use relevant literature for reference.

4.The actual content of this paper does not involve the effect of terrain factors such as slope and roughness on the soil organic matter. It is recommended that the future study may be explained in the final discussion section.

5.It is noted that your paper needs careful editing by someone with expertise in technical English editing paying particular attention to the whole text so that the goals and results are clear to the reader.

Author Response

Reviewer #3

Comment 1: This paper studies the rainfed vineyard in a typical Mediterranean climate region and the effects of cover crops on soil organic matter, which has certain value for land degradation and sustainable management. This research is suitable for publication in Land. However, this paper still needs some improvement.

Reply to R3.1: Thank you very much for these words. We appreciate them. We have addressed the required comments.

Comment 2: 1.In the abstract(P3L102-104 and P3L115-117), the authors described that the slope gradient also plays an important role in the effect of cover crops on soil organic matter. However, the relevant factors have not been analyzed in this paper. Please explain why.

Reply to R3.2: The topographic factors, such as the slope gradient, soil roughness, convergence index and index of connectivity were evaluated and analysed in detail in the previous studies done in Los Oncenos sub-catchment. We cited these studies and included the relevant information in the text. In this study, we prefer to be focused on the specific benefits of the ground coverage on the loss of organic matter.

Comment 3: 2.In the legend of Figure 1a on P6, DEM data should be marked with units in parentheses. In the legend of Figure 1b, the unit of drainage area is wrong. It is recommended to use full name or abbreviation for the scale unit of the drawings in the paper, and unify the case of the marked letters in the drawing and the letters in the drawing title.

Reply to R3.3: We have corrected the units in Figure 1A. The units of the drainage area is square meters (m2) as it appears in the legend of the Figure 1B. We have modified the figure captions to use capital letters following the style used in the figures.

Comment 4: 3.In L268-271 on P7, the authors found that the OM concentration in the sediment was positively correlated with the runoff yield and negatively correlated with the sediment yield. However, the author only made a data description. Please analyze the cause of this phenomenon combined with the field situation and use relevant literature for reference.

Reply to R3.4: Thank you for this remark. We agree with Reviewer #3 that this aspect of the hydrological response of the soil is very interesting. We have added a possible explanation and a new reference: “These results may be explained by the non-linear relationship that exists between runoff and sediment yield observed by different authors in distinct sites [44].” The new reference is: Detailed spatial analysis of SWAT-simulated surface runoff and sediment yield in a mountainous watershed in China. Hydrological Sciences Journal 2015, 60(5), 784–800. https://doi.org/10.1080/02626667.2014.965172

Comment 5: 4.The actual content of this paper does not involve the effect of terrain factors such as slope and roughness on the soil organic matter. It is recommended that the future study may be explained in the final discussion section.

Reply to R3.5: We appreciate this comment and have written the following further research line that is supported by a new reference: “…the spatial pattern of soil redistribution strongly depends on the magnitude and intensity of the rainfall event and clear changes can be found within the same site [43]. Therefore, collected runoff and sediment during the low intensity events may correspond to the drainage are located near the ST. Most likely, only during the heaviest storm events, eroded particles from any part of the drainage area reached the ST. Further research should be focus on identifying the travel distance of detached soil.” The new reference is: Severe soil erosion during a 3-day exceptional rainfall event: Combining modelling and field data for a fallow cereal field. Hydrological Processes 2015, 29(10), 2358–2372. https://doi.org/10.1002/hyp.10370

Comment 6: 5.It is noted that your paper needs careful editing by someone with expertise in technical English editing paying particular attention to the whole text so that the goals and results are clear to the reader.

Reply to R3.6: One of our colleagues, with proficiency level in English, has carefully checked the complete text in order to refine the style and grammar and correct any error.

Round 2

Reviewer 2 Report

Improvements are noted. A few issues remain: 

Comment 13: L59 runoff coefficient: explain what coefficient measures. Does it mean 7.4% of soil mass was lost or soil erosion occurs on 7.4% of the area?
Reply to R2.13: We have added the meaning of runoff coefficient: "ratio between the amount of runoff volume and rainfall depth per unit of area"

So, according to your reply, the runoff coefficient will have the following unit:

runoff volume (cm^3) / rainfall depth (cm) = cm^2

which is different from %;

Comment 21: L177 synthetic weather station was generated: How? Better give a brief description and provide references;
Reply to R2.21: We have added new information about the two weather stations used to generate the synthetic weather station: "During the test period, values of rainfall depth were obtained from two weather stations, called ‘Barbastro-Oficina del Regante’ (managed by the Regional Government of Aragon; records every 30 min) and ‘Barbastro-CHEbro’ (managed by the Ebro River Basin water authorities; records every 15 min), that are located 4.2 and 5.2 Km eastern from the study area, respectively. Ben-Salem et al. [7] observed minor differences between the records of the two stations, and then, generated a synthetic weather station."

Your reply explains "why" you need a synthetic weather station, but not "how" you generated it.

Comment 29: L287 "2.7 times": How did you arrive at this number?
Reply to R2.29: In the previous sentence we included the numbers that we used to calculate this ratio.

I noted the numbers in the previous sentence. However, those numbers do not give a relationship of "2.7 times". 

Author Response

Reviewer #2. Round 2.

Comment 1: Improvements are noted. A few issues remain.

Reply to R2.1: Thank you for your words. We have addressed the remaining issues.

Comment 2: Comment 13 (1st round): L59 runoff coefficient: explain what coefficient measures. Does it mean 7.4% of soil mass was lost or soil erosion occurs on 7.4% of the area? Reply to R2.13: We have added the meaning of runoff coefficient: "ratio between the amount of runoff volume and rainfall depth per unit of area"

Reply to R2.2: The runoff coefficient is a metric to estimate the ratio between the amount of water that leaves –via runoff– from a hydrological unit and the amount of water that enters –via rainfall– into the same unit. This metric is very common in hydrological studies at catchment or plot scales, but it is not a measurement of soil erosion. We have added this detailed explanation in the revised version.

Comment 3: So, according to your reply, the runoff coefficient will have the following unit: runoff volume (cm^3) / rainfall depth (cm) = cm^2 which is different from %;

Reply to R2.3: The amount of water that leaves from or enters into a hydrological unit is measured in mm or L m–2. Both units are correct because the density of water is 1 gr cm–3. We have added the units in the revised version.

Comment 4: Comment 21 (1st round): L177 synthetic weather station was generated: How? Better give a brief description and provide references; Reply to R2.21: We have added new information about the two weather stations used to generate the synthetic weather station: "During the test period, values of rainfall depth were obtained from two weather stations, called ‘Barbastro-Oficina del Regante’ (managed by the Regional Government of Aragon; records every 30 min) and ‘Barbastro-CHEbro’ (managed by the Ebro River Basin water authorities; records every 15 min), that are located 4.2 and 5.2 Km eastern from the study area, respectively. Ben-Salem et al. [7] observed minor differences between the records of the two stations, and then, generated a synthetic weather station." Your reply explains "why" you need a synthetic weather station, but not "how" you generated it.

Reply to R2.4: We have the following explanation: “Firstly, the temporal resolution of both records were equalized by calculating the values of precipitation every 30 min. Then, the mean value of precipitation was calculated and assigned this value to the synthetic weather station.”

Comment 5: Comment 29 (1st round): L287 "2.7 times": How did you arrive at this number? Reply to R2.29: In the previous sentence we included the numbers that we used to calculate this ratio. I noted the numbers in the previous sentence. However, those numbers do not give a relationship of "2.7 times".

Reply to R2.5: Thank you very much for this remark, we appreciate it. We have corrected the text as follows: “Despite the total amount of OM loss in the ST2 was 1.9 times higher than in the ST3, the differences between the two ST were not significant (P = 0.277). Based on the extent of the drainage area of the two STs, the average annual rate of OM loss was of 1.29 and 0.35 kgOM ha−1 yr−1 in the ST2 and ST3. Although the average annual rate of OM loss in the ST2 was 3.6 times higher than in the ST3, the differences between the two STs during the test period were not significant (P = 0.149).”